# Synthesis of Novel Aminothiazole Derivatives as Promising Antiviral, Antioxidant and Antibacterial Candidates

**DOI:** 10.3390/ijms23147688

**Published:** 2022-07-12

**Authors:** Rūta Minickaitė, Birutė Grybaitė, Rita Vaickelionienė, Povilas Kavaliauskas, Vidmantas Petraitis, Rūta Petraitienė, Ingrida Tumosienė, Ilona Jonuškienė, Vytautas Mickevičius

**Affiliations:** 1Department of Organic Chemistry, Faculty of Chemical Technology, Kaunas University of Technology, Radvilėnų pl. 19, LT-50254 Kaunas, Lithuania; ruta.minickaite@ktu.edu (R.M.); birute.grybaite@ktu.lt (B.G.); rita.vaickelioniene@ktu.lt (R.V.); povilas.kavaliauskas@ktu.edu (P.K.); ingrida.tumosiene@ktu.lt (I.T.); vytautas.mickevicius@ktu.lt (V.M.); 2Transplantation-Oncology Infectious Diseases Program, Division of Infectious Diseases, Department of Medicine, Weill Cornell Medicine of Cornell University, 527 East 68th Street, New York, NY 10065, USA; vip2007@med.cornell.edu (V.P.); rop2016@med.cornell.edu (R.P.); 3Department of Microbiology and Immunology, University of Maryland Baltimore School of Medicine, 655 W. Baltimore Street, Baltimore, MD 21201, USA; 4Institute of Infectious Diseases and Pathogenic Microbiology, Birštono Str. 38A, LT-59116 Prienai, Lithuania; 5Biological Research Center, Veterinary Academy, Lithuanian University of Health Sciences, Tilžės St. 18, LT-47181 Kaunas, Lithuania

**Keywords:** thiazole, antiviral, oxidative stress, antioxidant, antibacterial, bioactivity

## Abstract

It is well-known that thiazole derivatives are usually found in lead structures, which demonstrate a wide range of pharmacological effects. The aim of this research was to explore the antiviral, antioxidant, and antibacterial activities of novel, substituted thiazole compounds and to find potential agents that could have biological activities in one single biomolecule. A series of novel aminothiazoles were synthesized, and their biological activity was characterized. The obtained results were compared with those of the standard antiviral, antioxidant, antibacterial and anticancer agents. The compound bearing 4-cianophenyl substituent in the thiazole ring demonstrated the highest cytotoxic properties by decreasing the A549 viability to 87.2%. The compound bearing 4-trifluoromethylphenyl substituent in the thiazole ring showed significant antiviral activity against the PR8 influenza A strain, which was comparable to the oseltamivir and amantadine. Novel compounds with 4-chlorophenyl, 4-trifluoromethylphenyl, phenyl, 4-fluorophenyl, and 4-cianophenyl substituents in the thiazole ring demonstrated antioxidant activity by DPPH, reducing power, FRAP methods, and antibacterial activity against *Escherichia coli* and *Bacillus subtilis* bacteria. These data demonstrate that substituted aminothiazole derivatives are promising scaffolds for further optimization and development of new compounds with potential influenza A-targeted antiviral activity. Study results could demonstrate that structure optimization of novel aminothiazole compounds may be useful in the prevention of reactive oxygen species and developing new specifically targeted antioxidant and antibacterial agents.

## 1. Introduction

Progress in organic and medicinal chemistry allows for the design, synthesis and optimization of the structures of novel thiazole compounds. Thiazole compounds are widely used in the pharmaceutical industry for the design of therapeutics and novel biosensors for their antioxidant [1,2,3], antibacterial [4,5,6], anti-proliferative [7], antiparasitic [8], anti-inflammatory [9], analgesic [10], neuroprotective [11], antiviral (including SARS-CoV-2) [12], and anti-HIV activities [13]. Studies of thiazole-structure activities promote the improvement of their chemical/biochemical and therapeutic properties and anti-TB activity, mainly against resistant *Mycobacterium tuberculosis* (Mtb) strains [14].

Thiazole derivatives are important in designing and discovering pharmaceuticals, and they are incorporated into the structures of antimicrobial (acinitrazole and sulfathiazole) [15], antidepressant (pexole) [16], antineoplastic (bleomycin) [17], anti-HIV (ritonavir) [18], antiasthmatic (cinalukast) [19], antiulcer (nizatidine) [20], antibiotic (penicillin), thiamine (vitamin B1) [21], non-steroidal immunomodulatory (fanetizole) [22], anti-inflammatory (anetizole, meloxicam, fentiazac), analgesic, antineoplastic (tiazofurin, dasatinib), antifungal (ravuconazole), antiparasitic (nitazoxanide), anti-inflammatory (anetizole, meloxicam, fentiazac), and antiulcer (nizatidine) [23,24] drugs and agents. Thiazoles possess a polyoxygenated phenyl molecule that showed anti-fungal activity [25] and thiazolium also possesses bis-thiazolium salts that have been screened as potent antimalarial agents.

Novel thiazole compounds also participate in bioluminescent systems, which are focused on the substrate specificity of D-luciferin for luciferase. Previous studies have discovered that a thiazoline ring of the original structure should be conserved for emitting bioluminescence, whereas an aromatic ring and its substituents could be modified [26]. A thiazole ring containing firefly luciferin is responsible for the characteristic yellow-light emission from fireflies [27]. Bioluminescent systems have important roles in medicinal biology and clinical applications for the design of protein-based biosensors for detection of the SARS-CoV-2 virus [28].

Cancer is the leading disease and one of the significant healthcare challenges of the 21st century. It is important to mention that thiazole compounds could be used as highly versatile scaffolds for the development of anticancer agents [29,30]. A number of thiazole derivatives have been described to show potent anticancer activity by inhibiting tubulin polymerization [31,32]. Excessive reactive-oxygen species (ROS) production, oxidative stress, mitochondrial disfunction, and lipid peroxidation have also been implicated in cancer pathology. Oxidative stress is considered the starting point for tissues’ chronic inflammation and cancer [33].

The demand of novel antioxidant agents has been increasing because of the long-term safety and a negative consumer perception about synthetic antioxidants such as butylhydroxyanisole, BHA, and butylhydroxytoluene, BHT, which showed toxic and carcinogenic side effects in animal models [34]. The discovery of compounds that can have both antimicrobial and antioxidant activities with no toxic effects on health is, therefore, highly awaited [35].

Camalexin (3-thiazol-2-yl-indole) is an indole alkaloid phytoalexin, which is induced by phytopathogens and accumulates in various *Brassicaceae* plant organs or tissues. Many studies have characterized that camalexin and its derivatives have been found to possess significant anticancer, antifungal, antiviral, and antibacterial activities [36]. Camalexin biosynthesis is dependent on cysteine, and glutathione reduced form (GSH) is the direct precursor of the thiazole ring [37].

Our research interest was focused on the design, synthesis, screening, and investigation of novel aminothiazole compounds for their influenza A-targeted antiviral, antioxidant, and antibacterial potential. To counter increasing drug resistance, thiazoles could be considered as a promising scaffold to generate novel bioactive derivatives.

Here, we discuss the important challenges for the understanding of aminothiazoles’ activities and the current strategies for improving chemical, biological, and therapeutical characteristics, with the combination of three targets in one biomolecule structure.

## 2. Results

### 2.1. Synthesis of Novel Aminothiazoles

The starting compound **2** was synthesized from 4-aminoacetanilide (**1**) and acrylic acid. Initially, we used the method described in the literature [38]. Although the authors report a 77.4% yield, the reaction under the specified conditions yielded only 7% of the product. This led to the study of the above-mentioned reaction using different solvents. Reactions were carried out in toluene, 1,4-dioxane, 2-propanol, and tetrahydrofuran (THF) at reflux for 24 h. The results are shown in Table 1.

The data demonstrate that a reaction in THF allowed the preparation of compound **2**, providing the highest yield of 70%. Triplets at 2.47 (CH_2_CO) and 3.20 (NHCH_2_), the singlet at 3.77 (NHCH_2_), and a broad singlet at 11.81 (COOH) in ^1^H NMR for the compound confirmed the formation of the *β-*alanine fragment in the structure.

The cyclization of *β*-alanine moiety to 2-thioxotetrahydropyrimidinedione **3** was performed by refluxing carboxylic acid **2** with potassium thiocyanate in acetic acid. Compound **3** was separated by dilution of the reaction mixture with water. The obtained product was applied for the preparation of thioureido acid **4** (Figure 1). For this purpose, compound **3** was dissolved in hot, aqueous 5% sodium hydroxide, and the obtained solution of the carboxylic acid sodium salt was filtered off and transferred to an acidic form by acidifying the filtrate with acetic acid to pH 6.

The prepared thioureido acid **4** was applied for the synthesis of thiazoles **5a**–**e**. The products **5a**–**e** have been achieved by a reaction of thioureido acid **4** and the corresponding bromoacetophenone in water, with the presence of sodium carbonate in the reaction mixture. The ^1^H NMR spectra of these compound singlets, in the range of 7.07–7.42 ppm (^13^C, 102.09–106.41 ppm), and the additional peaks in the aromatic region, confirm the presence of a 4-arylthiazole moiety.

The next goal of this study was the transformation of the acetamide fragment to an amine group. Reactions occurred easily and rapidly and were completed after refluxing in aqueous 5% hydrochloric acid for 1 h. Products were isolated by neutralizing the reaction mixtures with sodium acetate to pH 6. By careful assignment of the peaks in the ^1^H and ^13^C NMR spectra, the structures **6** were elucidated. Comparison of the spectra of compounds **5** and **6** showed obvious differences, i.e., the spectra of the latter compounds do not contain singlets of the methyl group of the acetamide moiety, but broad singlets of amino group are visible at approximately 5.35 ppm. In addition, the additional proof of the new structure is the absence of the resonance of the carbonyl carbon of the CH_3_CO fragment in the ^13^C NMR spectra of compounds **6**, which are clearly visible in the analogous spectra of derivatives **5** at approximately 168.45 ppm.

To obtain thiazolone derivative **7**, a ring-closure reaction was carried out where thioureido acid **4** was reacted with monochloroacetic acid in an aqueous sodium carbonate solution, where acidification to pH 6 after completion of the reaction produced the target compound **7** (Figure 2). The structure of 7 was approved based on the data of elemental analysis and NMR as well as IR spectroscopy.

The condensation of the obtained thiazolone **7** with various aromatic aldehydes was performed under analogous conditions, such as synthesis, instead of the monochloroacetic acid using the corresponding aromatic aldehydes. The reactions afforded 5-[(substituted phenyl)methylidene] thiazolones **8a**–**d**. The spectra of compounds **8** do not contain the proton singlet of the SCH_2_, which arises at 3.91 (^1^H) and 40.59 (^13^C) ppm in the NMR spectra for compound **7**. In the structure of compounds **8**, **a** = CHAr moiety is attached to the 5-position of the thiazole ring. The NMR spectra of formed molecules **8** show obvious differences in comparison with the NMR spectra of **7**. The increase in spectral lines in the aromatic region of the ^1^H (in the interval of 7.59–7.61 ppm for CH and in the range of 7.21–7.63 ppm for H_Ar_) and ^13^C (observed in the interval of 129–134 ppm) NMR spectra correspond to the number of hydrogen and carbon atoms of the new fragment. Then, 3-((4-aminophenyl)(4-(4-chlorobenzylidene)-5-oxo-4,5-dihydrothiazol-2-yl)amino)propanoic acid (**9**) was obtained by the deacetylation of compound **8c** in the aqueous 5% hydrochloric-acid solution. The formed amino group was confirmed by a broad singlet, which was visible at 5.61 ppm.

Next, 2-thioxotetrahydropyrimidinedione **3** was used to prepare derivative **10** with an amino group in its structure. For this reason, compound **3** was refluxed in an aqueous 5% HCl solution. Product **10** was isolated by neutralizing the reaction mixture with sodium acetate to pH 6.

The obtained product **10** was used for the preparation of thioureido acid **11** (Figure 3). For this purpose, compound **10** was dissolved in a hot, aqueous 5% sodium hydroxide solution, then filtered off and acidified with acetic acid to pH 6, to transfer sodium salt to the acidic form. Comparison of the spectra of compounds **3** and **10** showed that the singlet of the methyl group of the acetamide moiety is absent, but a broad singlet of an amino group is visible at 5.20 ppm.

### 2.2. Study of Cytotoxic Activity of Compounds **3**–**11**

To explore the in vitro cytotoxicity of compounds **3**–**11**, we used A549 human pulmonary endothelial cells and a MTT viability assay. We exposed the cells to the fixed concentration of 100 µM of each compound for 48 h and subsequently measured the viability. Compounds **3**–**11** demonstrated overall favorable properties with low cytotoxicity. Among all tested compounds, **6d**, bearing 4-cianophenyl substituent in the thiazole ring, demonstrated the highest cytotoxic properties by decreasing the A549 viability to 87.2%. All compounds failed to reduce the A549 viability by 50%, suggesting good in vitro safety profiles (Figure 1).

### 2.3. Study of Antiviral Activity of Compounds **3**–**11**

To explore the potential antiviral ability of synthesized compounds, we used an MDCK influenza in vitro infection model [39,40]. Prior to infection, we pretreated the MDCK cells with 100 µM of each compound, or oseltamivir and amantadine that served as antiviral controls. Compounds **3**–**11** showed structure-dependent antiviral activity against influenza A/Puerto Rico/8/34 H1N1 strain and were able to significantly (*p* < 0.05) restore MDCK viability in comparison to the untreated control (UC) (Figure 2). Compounds **8d**, **5e**, **5d**, and **6e** showed the highest antiviral activity (*p <* 0.001) in comparison to UC. The antiviral activity of compounds **8d**, **5e**, **5d**, and **6e** at 100 µM was similar or greater than oseltamivir and amantadine. Furthermore, compound **5e**, bearing 4-trifluoromethylphenyl substituent in the thiazole ring, showed significantly higher antiviral activity (*p* < 0.0162) than oseltamivir (Figure 2).

These data demonstrate that substituted aminothiazole derivatives are promising scaffolds for further optimization and the development of new compounds with potential influenza A-targeted antiviral activity.

### 2.4. Measurement of Antioxidant Activities

Ferric ion (Fe^3+^) reduces the antioxidant power test (ferricyanide/Prussian blue assay). Bioactive compounds with antioxidant-reducing activity transfer an electron to ferricyanide’s Perls Prussian blue complex, reducing Fe[(CN)_6_]_3_ to Fe[(CN)_6_]_2_ [41,42]. Increasing absorbance at 700 nm shows an increase in the reductive ability of the reaction complex [43].

The data obtained from the study (Figure 3) demonstrated that the compounds bearing 4-cianophenyl **6d**, 4-chlorophenyl **6c**, and 4-trifluoromethylphenyl **6e** substituents in the thiazole ring showed the highest ferric ion (Fe^3+^)-reducing power. The scaffolds of the compounds bearing 5-benzylidene **8a**, {5-[(4-bromophenyl)methylidene] **8d**, 5-[(4-chlorophenyl)methylidene] **8c**, 5-[(4-fluorophenyl)methylidene] **8b**, and 4-oxo-4,5-dihydro-1,3-thiazol-2-yl)amino] **7** in analyzed propanoic acid derivatives exhibited the lowest reducing antioxidant power.

Ferric-reducing antioxidant power (FRAP assay). The ferric ion reducing antioxidant power (FRAP)-reaction mechanism is focused on electron transfer (ET)-based bioassays. The FRAP technique is carried out at pH 3.6 and evaluates the reduction in ferric Fe^3+^ complex of 2,4,6-tripyridyl-*s*-triazine Fe(TPTZ)^3+^ to the blue-colored ferrous (Fe^2+^) complex Fe(TPTZ)^2+^ by bioactive compounds.

Measuring the increasing absorption at 593 nm using a spectrophotometer monitors this reduction, and results are expressed as a Fe^2+^ µmol/L concentration. [44].

The results (Figure 4) of the study exhibited that compounds bearing 4-chlorophenyl **6c** (123.20 µmol/L), phenyl **6a** (114.18 µmol/L), 4-fluorophenyl **6b** (111.83 µmol/L), and 4-trifluoromethylphenyl **6e** (106.53 µmol/L) substituents in the thiazole ring showed the highest reduced FRAP power, in comparison with BHT (67.73 µmol/L). The compounds **8d**, **8c**, **8a**, **7**, and **8b** demonstrated the lowest FRAP power.

DPPH radical-scavenging assay. The 1,1-Diphenyl-2-picrylhydrazyl (DPPH•) radical-scavenging assay has been one of the most commonly applied methods to determine antioxidant activity [45,46]. The DPPH test is based on either a hydrogen-atom transfer (HAT) or a single-electron transfer (SET) mechanism. Bioactive compounds are able to donate a hydrogen atom to reduce the stable DPPH• radical (a deep purple color) to the yellow-colored non-radical compound at 517 nm.

As seen from the results presented in Figure 5, the compounds bearing 4-fluorophenyl **6b** (83.63%) and 4-phenyl **6a** (52.04%) substituents in the thiazole ring, as well as 3-(1-(4-aminophenyl)thioureido)propanoic acid **11** (66.5%) and 3-[(4-acetamidophenyl)(carbamothioyl)amino]propanoic acid **4** (49.36%), possess a very high DPPH radical-scavenging ability in comparison with BHT (45.14%). The compounds **7, 8a,** and **8d** demonstrated low DPPH inhibition. The compounds **5a**–**5e** did not show antioxidant activity according to the DPPH method.

### 2.5. Evaluation of Antibacterial Activity

The novel derivatives were screened for their in vitro antibacterial activity against *Escherichia coli* (Gram-negative) and *Bacillus subtilis* (Gram-positive) bacteria strains. Antimicrobial tests were conducted using the agar well-diffusion method. Ciprofloxacin was used as the reference antibiotic for the in vitro antibacterial activity.

The results of the antibacterial study against *E. coli* (Figure 6) illustrated that the compounds bearing 4-cianophenyl **6d**, 4-fluorophenyl **6b**, 4-chlorophenyl **8c**, 4-trifluoromethylphenyl **5e**, and phenyl **6a** substituents in the thiazole ring showed the highest antibacterial activity against *E. coli*. The compounds **8b**, **8a**, **7**, **3**, **4**, **10**, and **11** showed the lowest antibacterial activity against *E. coli*.

The results of the antibacterial study against *B. subtilis* demonstrated (Figure 7) that the compounds bearing 4-cianophenyl **6d**, 4-fluorophenyl **6b**, 4-fluoro **5b**, 4-chlorophenyl **8c**, 4-trifluoromethylphenyl **5e**, and phenyl **6a** and **5a** substituents in the thiazole ring showed the highest antibacterial activity against this strain. The compounds **8a**, **8b**, **3**, **7**, **10**, **4,** and **11** showed the lowest antibacterial activity against *B. subtilis.*

## 3. Discussion

It is important to develop an understanding of these novel aminothiazole compounds by integrating their synthesis and biological activities (antiviral, antioxidant, antibacterial). The present study results demonstrated that the compound 3-{(4-aminophenyl)[4-(4-cianophenyl)-1,3-thiazol-2-yl]amino}propanoic acid (**6d**) showed the highest cytotoxic properties. Screening of the tested compounds for antiviral activity has revealed that the compounds 3-((4-aminophenyl){4-[4-(trifluoromethyl)phenyl]thiazol-2-yl}amino)propanoic acid (**6e**) and 3-{(4-acetamidophenyl)[4-(4-trifluoromethylphenyl)-1,3-thiazol-2-yl]amino}propanoic acid (**5e**) exhibited significant influenza A-targeted antiviral activity, in comparison with oseltamivir and amantadine. The principle of antioxidant activity is focused on the availability of electrons to neutralize any free radicals. Moreover, antioxidant activity is related to the nature of the hydroxylation pattern on the aromatic ring. Various antioxidant methods have been described to evaluate antioxidant properties of pharmaceuticals, antioxidants, and other bioactive samples [43]. The antioxidant bioassays should be based on the elucidation of the structure–antioxidant-activity relationship of the bioactive molecules. Screening of the reduction power of the bioactive compounds gives information not only about their reducing ability but also reveals their thermodynamic parameters. It was determined that thiazole compounds bearing 4-cyanophenyl **6d**, 4-chlorophenyl **6c**, and 4-trifluoromethylphenyl **6e** substituents in the thiazole ring showed the highest **6d** > **6c** > **6e** > **BHT** > **11** > **6a** > **6b** > **5c** antioxidant-reducing activity. The compounds **8a**, **8d**, **8c**, **8b**, and **7** demonstrated the lowest antioxidant-reducing power. It could be concluded that aminothiazole compounds bearing 4-chlorophenyl **6c**, 4-trifluoromethylphenyl **6e**, phenyl **6a**, 4-fluorophenyl **6b**, and 4-cianophenyl **6d** substituents in the thiazole ring exhibited the highest **6c** > **6a** > **6e** > **6b** > **6d** > **11** > **5a** > **5d** > **5b** > **BHT** reduced FRAP power. The model of scavenging the stable DPPH radical is a widely used method to evaluate the free-radical-scavenging ability of various samples. It is a stable nitrogen-centered free radical, the color of which changes from violet to yellow upon reduction by either the process of hydrogen or electron donation. It was determined that the thiazole compounds **6b** > **11** > **6a** > **4** > **BHT** showed the highest antioxidant activity by DPPH assay. The acquired results of the antibacterial activity showed that compounds bearing 4-cianophenyl **6d**, 4-fluorophenyl **6b**, 4-chlorophenyl **8c**, 4-trifluoromethylphenyl **5e**, **and** phenyl **6a** substituents in the thiazole ring have been indicated as the most active antibacterial agents against *E. coli*. The compounds **6d** = **6a** = **6b** = **8c** = **5a** = **5b** = **5e** > **5d** = **5c** = **8d** showed the highest antibacterial activity against *B. subtilis*. The antibacterial activity of the novel compounds could be related with the existence of electron-withdrawing groups in the thiazole ring.

## 4. Materials and Methods

### 4.1. Synthesis of Novel Compounds

Melting points were determined on a MEL-TEMP (Electrothermal, A Bibby Scientific Company, Burlington, NJ, USA) melting point apparatus and are uncorrected. FT-IR spectra (ν, cm^−1^) were recorded on a Perkin–Elmer Spectrum BX FT–IR spectrometer (Perkin–Elmer Inc., Waltham, MA, USA) using KBr pellets. ^1^H and ^13^C-NMR spectra were recorded in DMSO-d_6_ on a Bruker Avance III (400 MHz, 101 MHz and 700 MHz, 176 MHz) spectrometer. Chemical shifts (δ) are reported in parts per million (ppm) calibrated from TMS (0 ppm) as an internal standard for ^1^H-NMR and DMSO-*d_6_* (39.43 ppm) for ^13^C-NMR. The reaction course and purity of the synthesized compounds was monitored by TLC using aluminum plates coated with silica gel 60 F254 (MerckKGaA, Darmstadt, Germany). Reagents were obtained from Sigma-Aldrich (St. Louis, MO, USA).

*N*-[4-(acetylamino)phenyl]-*β*-alanine (**2**)

A mixture of 4-aminoacetanilide (**1**) (0.3 mol, 45 g), acrylic acid (0.45 mol, 30.88 mL), and THF (90 mL) was heated at reflux for 24 h. After completion of the reaction (TLC), it was cooled to room temperature, and the formed crystalline solid was filtered off and washed with THF.

Yield 47 g (70%). Melting point coincides with that given in the literature [38].

IR (KBr), ν, cm^−1^: 2670 (OH), 3320 (NH), 1718, 1590 (2 C=O).

^1^H NMR (400 MHz, DMSO-*d_6_*) δ: 1.96 (s, 3H, CH_3_), 2.47 (t, *J* = 6.9 Hz, 2H, CH_2_CO), 3.20 (t, *J* = 8.7 Hz, 2H, NHC*H_2_*), 3.77 (s, 1H, N*H*CH_2_), 6.50 (d, *J* = 8.7 Hz, 2H, H_Ar_), 7.27 (d, *J* = 8.7 Hz, 2H, H_Ar_), 9.51 (s, 1H, NHCO), 11.81 (br s, 1H, COOH).

^13^C NMR (101 MHz, DMSO-*d_6_*) δ: 23.72 (CH_3_), 33.83 (*C*H_2_CO), 39.21 (NHCH_2_), 112.04, 120.91, 128.81, 144.75, 167.28, 173.31 (C_Ar_, CH_3_*C*O, COOH).

Anal. Calcd. for C_11_H_14_N_2_O_3_, %: C 59.45; H 6.35; N 12.60. Found, %: C 59.56; H 6.31; N 12.71.

*N*-[4-(tetrahydro-4-oxo-2-thioxo-1(2*H*)-pyrimidinyl)phenyl]acetamide (**3**)

A mixture of *β*-alanine (**2**) (10 g, 45 mmol), potassium thiocyanate (13.10 g, 135 mmol), and acetic acid (25 mL) was heated at reflux for 24 h then cooled to room temperature and diluted with 100 mL of water. The formed precipitate was filtered off and washed with water. Yield 8.53 g (72%), m. p. 227−228 °C (from ethanol).

IR (KBr), ν, cm^−1^: 3401−3115 (2 NH), 1683, 1598 (2 C=O).

^1^H NMR (400 MHz, DMSO-*d_6_*) δ: 2.05 (s, 3H, CH_3_), 2.79 (t, *J* = 6.9 Hz, 2H, CH_2_CO), 3.87 (t, *J* = 6.9 Hz, 2H, NCH_2_), 7.25 (d, *J* = 8.6 Hz, 2H, H_Ar_), 7.60 (d, *J* = 8.6 Hz, 2H, H_Ar_), 10.05 (s, 1H, NH), 11.20 (s, 1H, NH).

^13^C NMR (101 MHz, DMSO-*d_6_*) δ: 23.99 (CH_3_), 30.43 (*C*H_2_CO), 48.88 (NCH_2_), 119.36, 127.27, 138.35, 139.93, 166.97, 168.37, 179.40 (C_Ar_, CH_3_*C*O, *C*OCH_2_, C=S).

Anal. Calcd. for C_12_H_13_N_3_O_2_S, %: C 54.74; H 4.98; N 15.96. Found, %: C 54.84; H 5.03; N 15.84.

3-[(4-Acetamidophenyl)(carbamothioyl)amino]propanoic acid (**4**)

Compound **3** (7.90 g, 30 mmol) was dissolved in hot, aqueous 5% sodium hydroxide (22 mL), and the obtained solution was filtered off. After cooling to room temperature, the solution was acidified with acetic acid to pH 6. The formed solid was filtered off and washed with water.

Yield 7.93 g (94%), m.p. 137−138 °C (from ethanol).

IR (KBr), ν, cm^−1^: 3404−3167 (NH, OH), 1657, 1596 (2 C=O).

^1^H NMR (400 MHz, DMSO-*d_6_*) δ: 2.06 (s, 3H, CH_3_), 2.53 (t, *J* = 8.1 Hz, 2H, CH_2_CO), 4.15 (t, *J* = 7.8 Hz, 2H, NCH_2_), 6.40 (br. s, 2H, NH_2_CS), 7.14 (d, *J* = 8.4 Hz, 2H, H_Ar_), 7.66 (d, *J* = 8.4 Hz, 2H, H_Ar_), 10.14 (s, 1H, NH), 12.22 (br. s, 1H, COOH).

^13^C NMR (101 MHz, DMSO-*d_6_*) δ: 24.02 (CH_3_), 32.21 (*C*H_2_CO), 50.38 (NCH_2_), 120.04, 128.08, 136.21, 138.98, 168.45, 172.46, 181.68 (C_Ar_, CH_3_*C*O, COOH, C=S).

Anal. Calcd. for C_12_H_15_N_3_O_3_S, %: C 51.23; H 5.37; N 14.94. Found, %: C 51.38; H 5.47; N 14.77.

General procedure for the preparation of compounds **5a**–**e.**

To a solution of thioureido acid **4** (0.56 g, 2 mmol) and Na_2_CO_3_ (0.64 g, 6 mmol) in water (10 mL), the corresponding bromoacetophenone (2.4 mmol) was added, and the mixture was heated at reflux for 3 h. After completion of the reaction (TLC), the hot reaction mixture was filtered off, and the filtrate was acidified with acetic acid to pH 6. The formed precipitate was filtered off, washed with water, and purified by dissolving it in aqueous Na_2_CO_3_ solution (5 g, 40 mL of water), before filtering and acidifying the filtrate with acetic acid to pH 6.

3-[(4-Acetamidophenyl)(4-phenyl-1,3-thiazol-2-yl)amino]propanoic acid (**5a**)

Yield 0.53 g (69%), m. p. 132–133 °C.

IR (KBr), ν, cm^−1^: 3305–3129 (NH, OH), 1694, 1632 (2 C=O).

^1^H NMR (400 MHz, DMSO-*d_6_*) δ: 2.07 (s, 3H, CH_3_), 2.58 (t, *J* = 7.3 Hz, 2H, CH_2_CO), 4.12 (t, *J* = 7.5 Hz, 2H, NCH_2_), 7.10 (s, 1H, H_Ar_), 7.28 (t, *J* = 7.3 Hz, 1H, H_Ar_), 7.31–7.47 (m, 4H, H_Ar_), 7.68 (d, *J* = 8.7 Hz, 2H, H_Ar_); 7.86 (d, *J* = 7.8 Hz, 2H, H_Ar_), 10.14 (s, 1H, NH).

^13^C NMR (101 MHz, DMSO-*d_6_*) δ: 24.01 (CH_3_), 33.27 (*C*H_2_CO), 49.10 (NCH_2_), 102.49, 120.21, 125.66, 127.46, 127.63, 128.53, 134.73, 138.50, 139.28, 150.40, 168.43, 169.29, 173.16 (C_Ar_, CH_3_*C*O, COOH).

Anal. Calcd. For C_20_H_19_N_3_O_3_S, %: C 62.98; H 5.02; N 11.02. Found, %: C 63.05; H 5.19; N 11.10.

3-{(4-Acetamidophenyl)[4-(4-fluorophenyl)-1,3-thiazol-2-yl]amino}propanoic acid (**5b**)

Yield 0.52 g (65%), m. p. 125–126 °C.

IR (KBr), ν, cm-1: 3419–3112 (NH, OH), 1666, 1598 (2 C=O).

1H NMR (400 MHz, DMSO-*d_6_*) δ: 2.06 (s, 3H, CH_3_), 2.50–2.57 (signal of the CH_2_CO (2H) overlaps with the DMSO-*d_6_*), 4.09 (t, J = 7.4 Hz, 2H, NCH_2_), 7.07 (s, 1H, H_Ar_), 7.21 (t, J = 8.7 Hz, 2H, H_Ar_), 7.36 (d, J = 8.5 Hz, 2H, H_Ar_), 7.68 (d, J = 8.5 Hz, 2H, H_Ar_), 7.89 (dd, J = 8.0, 5.9 Hz, 2H, H_Ar_), 10.19 (s, 1H, NH).

13C NMR (101 MHz, DMSO-*d_6_*) δ: 24.00 (CH_3_), 33.94 (CH_2_CO), 49.59 (NCH_2_), 102.09, 115.24, 115.45, 120.21, 127.59, 127.67, 131.36, 138.47, 139.30, 149.35, 160.36, 162.78, 168.45, 169.40; 173.54 (C_Ar_, CH_3_CO, COOH).

Anal. Calcd. For C_20_H_18_FN_3_O_3_S, %: C 60.14; H 4.54; N 10.52. Found, %: C 60.26; H 4.47; N 10.43.

3-{(4-Acetamidophenyl)[4-(4-chlorophenyl)-1,3-thiazol-2-yl]amino}propanoic acid (**5c**)

Yield 0.52 g (63%), m. p. 194–195 °C.

IR (KBr), ν, cm^−1^: 3294–3054 (NH, OH), 1672, 1600 (2 C=O).

^1^H NMR (400 MHz, DMSO-*d_6_*) δ: 2.07 (s, 3H, CH_3_), 2.66 (t, *J* = 7.2 Hz, 2H, CH_2_CO), 4.15 (t, *J* = 7.2 Hz, 2H, NCH_2_), 7.19 (s, 1H, H_Ar_), 7.37 (d, *J* = 8.7 Hz, 2H, H_Ar_), 7.45 (d, *J* = 8.5 Hz, 2H, H_Ar_), 7.69 (d, *J* = 8.7 Hz, 2H, H_Ar_), 7.87 (d, *J* = 8.5 Hz, 2H, H_Ar_), 10,13 (s, 1H, NH).

^13^C NMR (101 MHz, DMSO-*d_6_*) δ: 24.02 (CH_3_), 32.35 (CH_2_CO), 48.50 (NCH_2_), 103.50, 120.28, 127.37, 127.75, 128.58, 128.88, 129.55, 131.93, 138.73, 139.00, 168.48, 169.55, 172.64 (C_Ar_, CH_3_*C*O, COOH).

Anal. Calcd. For C_20_H_18_ClN_3_O_3_S, %: C 57.76; H 4.36; N 10.10. Found, %: C 57.57; H 4.21; N 10.22.

3-{(4-Acetamidophenyl)[4-(4-cianophenyl)-1,3-thiazol-2-yl]amino}propanoic acid (**5d**)

Yield 0.66 g (81%). M. p. 181–182 °C.

IR (KBr), ν, cm^−1^: 3332–2972 (NH, OH), 1688, 1605 (2 C=O).

^1^H NMR (400 MHz, DMSO-*d_6_*) δ: 2.07 (s, 3H, CH_3_), 2.62 (t, *J* = 7.3 Hz, 2H, CH_2_CO), 4.14 (t, *J* = 7.3 Hz, 2H, NCH_2_), 7.37 (d, *J* = 8.7 Hz, 2H, H_Ar_), 7.42 (s, 1H, H_Ar_), 7.69 (d, *J* = 8.7 Hz, 2H, H_Ar_), 7.85 (d, *J* = 8.3 Hz, 2H, H_Ar_), 8.04 (d, *J* = 8.3 Hz, 2H, H_Ar_), 10.13 (s, 1H, NH).

^13^C NMR (101 MHz, DMSO-*d_6_*) δ: 24.02 (CH_3_), 32.82 (*C*H_2_CO), 48.88 (NCH_2_), 106.41, 109.51, 119.07, 120.28, 126.24, 127.74, 132.64, 138.74, 138.78, 138.97, 148.65, 168.48, 169.65, 172.87 (C_Ar_, CH_3_*C*O, COOH).

Anal. Calcd. For C_21_H_18_N_4_O_3_S, %: C 62.05; H 4.46; N 13.78. Found, %: C 61.97; H 4.53; N 13.57.

3-{(4-Acetamidophenyl)[4-(4-trifluoromethylphenyl)-1,3-thiazol-2-yl]amino}propanoic acid (**5e**)

Yield 0.84 g (94%), m. p. 205–206 °C.

IR (KBr), ν, cm^−1^: 3407–3168 (NH, OH), 1741, 1707 (2 C=O).

^1^H NMR (400 MHz, DMSO-*d_6_*) δ: 2.07 (s, 3H, CH_3_); 2.41 (t, *J* = 7.9 Hz, 2H, CH_2_CO), 4,09 (t, *J* = 7.9 Hz, 2H, NCH_2_), 7.32 (s, 1H, H_Ar_), 7.36 (d, *J* = 8.6 Hz, 2H, H_Ar_), 7.70 (d, *J* = 8.6 Hz, 2H, H_Ar_), 7.73 (d, *J* = 8.3 Hz, 2H, H_Ar_), 8.06 (d, *J* = 8.3 Hz, 2H, H_Ar_), 10.35 (s, 1H, NH).

^13^C NMR (101 MHz, DMSO-*d_6_*) δ: 23.99 (CH_3_), 35.09 (CH_2_CO), 50.32 (NCH_2_), 104.94, 120.24, 123.06, 125.53, 126.15, 127.59, 138.50, 138.55, 139.25, 148.88, 168.50, 169.66, 173.68, 174.42 (C_Ar_, CH_3_*C*O, COOH).

Anal. Calcd. For C_21_H_18_N_3_F_3_O_3_S, %: C 56.12; H 4.04; N 9.35. Found, %: C 56.04; H 4.09; N 9.31.

General procedure for the preparation of derivatives **6a**–**e**

A solution of the corresponding compound **5****a**–**e** (1.5 mmol) in aqueous 5% hydrochloric acid (20 mL) was refluxed for 1 h, cooled down, and evaporated under reduced pressure; then, the residue was dissolved in water, and the solution was neutralized with sodium acetate to pH 6. The formed precipitate was filtered off and recrystallized from 2-propanol.

3-[(4-Aminophenyl)(4-phenylthiazol-2-yl)amino]propanoic acid (**6a**)

Yield 0.45 g (88%), m. p. 194–195 °C.

IR (KBr), ν, cm^−1^: 3294–3054 (NH, OH), 1672, 1600 (2 C=O).

^1^H NMR (400 MHz, DMSO-*d_6_*) δ: 2.63 (t, *J* = 6.5 Hz, 2H, CH_2_CO), 4.08 (t, *J* = 7.0 Hz, 2H, NCH_2_), 5.36 (br. s, 2H, NH_2_), 6.63 (d, *J* = 8.2 Hz, 2H, H_Ar_), 6.99–7.11 (m, 3H, H_Ar_), 7.27–7.42 (m, 3H, H_Ar_), 7.85 (d, *J* = 7.5 Hz, 2H, H_Ar_), 12.22 (br. s, 1H, OH).

^13^C NMR (101 MHz, DMSO-*d_6_*) δ: 32.41 (CH_2_CO), 48.41 (NCH_2_), 102.38, 114.68, 125.61, 127.37, 128.50, 128.59, 132.60, 134.88, 148.64, 150.41, 170.64, 172.78 (C_Ar_, COOH).

Anal. Calcd. for C_18_H_17_N_3_O_2_S, %: C 63.70; H 5.05; N 12.38. Found, %: C 57.57; H 4.21; N 10.22.

3-{(4-Aminophenyl)[4-(4-fluorophenyl)-1,3-thiazol-2-yl]amino}propanoic acid (**6b**)

Yield 0.46 g (86%), m. p. 194–195 °C.

IR (KBr), ν, cm^−1^: 3294–3054 (NH, OH); 1672, 1600 (2 C=O).

^1^H NMR (400 MHz, DMSO-*d_6_*) δ: 2.62 (t, *J* = 7.3 Hz, 2H, CH_2_CO), 4.06 (t, *J* = 7.3 Hz, 2H, NCH_2_), 5.33 (br. s, 2H, NH_2_), 6.62 (d, *J* = 8.5 Hz, 2H, H_Ar_), 7.00–7.07 (m, 3H, H_Ar_), 7.21 (t, *J* = 8.8 Hz, 2H, H_Ar_), 7.88 (dd, *J* = 8.4, 5.8 Hz, 2H, H_Ar_), 12.22 (br. s, 1H, OH).

^13^C NMR (101 MHz, DMSO-*d_6_*) δ: 32.40 (*C*H_2_CO), 48.41 (NCH_2_), 102.11, 114.67, 115.22, 15.43, 127.52, 127.60, 128.57, 131.50, 132.53, 148.66, 149.34, 160.31, 162.73, 170.73, 172.75 (C_Ar_, COOH).

Anal. Calcd. for C_18_H_16_FN_3_O_2_S, %: C 60.49; H 4.51; N 11.76. Found, %: C 57.57; H 4.21; N 10.22.

3-{(4-Aminophenyl)[4-(4-chlorophenyl)-1,3-thiazol-2-yl]amino}propanoic acid (**6c**)

Yield 0.46 g (82%), m. p. 171–172 °C.

IR (KBr), ν, cm^−1^: 3368–1974 (NH, OH), 1698 (C=O).

^1^H NMR (400 MHz, DMSO-*d_6_*) δ: 2.44–2.49 (signal overlaps with the DMSO-d_6_, 2H, CH_2_CO), 4.01 (t, *J* = 7.3 Hz, 2H, NCH_2_), 5,34 (br s, 2H, NH_2_), 6.62 (d, *J* = 8.3 Hz, 2H, H_Ar_), 7.01 (d, *J* = 8.3 Hz, 2H, H_Ar_), 7.08 (s, 1H, CH), 7.43 (d, *J* = 8.4 Hz, 2H, H_Ar_), 7.86 (d, *J* = 8.4 Hz, 2H, H_Ar_).

^13^C NMR (101 MHz, DMSO-*d_6_*) δ: 33.91 (*C*H_2_CO), 49.37 (NCH_2_), 102.82, 114.65, 127.28, 128.48; 128.54, 131.64, 132.71, 133.78, 148.53, 149.14, 170.75 (C_Ar_, COOH).

Anal. Calcd. for C_18_H_16_ClN_3_O_2_S, %: C 57.83; H 4.31; N 11.24. Found, %: C 57.71; H 4.27; N 11.21.

3-{(4-Aminophenyl)[4-(4-cianophenyl)-1,3-thiazol-2-yl]amino}propanoic acid (**6d**)

Yield 0.44 g (81%), m. p. 171–172 °C.

IR (KBr), ν, cm^−1^: 3368–1974 (NH, OH), 1698 (C=O).

^1^H NMR (400 MHz, DMSO-*d_6_*) δ: 2.62 (t, *J* = 6.6 Hz, 2H, CH_2_CO), 4.08 (t, *J* = 6.6 Hz, 2H, NCH_2_), 5,37 (br s, 2H, NH_2_), 6.63 (d, *J* = 8.0 Hz, 2H, H_Ar_), 7.04 (d, *J* = 8.0 Hz, 2H, H_Ar_), 7.35 (s, 1H, CH), 7.84 (d, *J* = 7.8 Hz, 2H, H_Ar_), 8.03 (d, *J* = 7.8 Hz, 2H, H_Ar_), 12.26 (br. s, 1H, OH).

^13^C NMR (101 MHz, DMSO-*d_6_*) δ: 32.36 (*C*H_2_CO), 48.44 (NCH_2_), 106.24, 109.36, 114.69, 119.08, 126.17, 128.56, 132.32, 132.59, 138.93, 148.65, 148.76, 170.93, 172.73 (C_Ar_, COOH).

Anal. Calcd. for C_19_H_16_N_4_O_2_S, %: C 62.62; H 4.43; N 15.37. Found, %: C 57.71; H 4.27; N 11.21.

3-((4-Aminophenyl){4-[4-(trifluoromethyl)phenyl]thiazol-2-yl}amino)propanoic acid (**6e**)

Yield 0.55 g (90%), m. p. 194–195 °C.

IR (KBr), ν, cm^−1^: 3294–3054 (NH, OH); 1672, 1600 (2 C=O).

^1^H NMR (400 MHz, DMSO-*d_6_*) δ: 2.62 (t, *J* = 7.3 Hz, 2H, CH_2_CO), 4.09 (t, *J* = 7.3 Hz, 2H, NCH_2_), 5.36 (br. s, 2H, NH_2_), 6.63 (d, *J* = 8.4 Hz, 2H, H_Ar_), 7.04 (d, *J* = 8.4 Hz, 2H, H_Ar_), 7.28 (s, 1H, CH), 7.74 (d, *J* = 8.2 Hz, 2H, H_Ar_), 8.06 (d, *J* = 8.2 Hz, 2H, H_Ar_), 12.27 (br. s, 1H, OH).

^13^C NMR (101 MHz, DMSO-*d_6_*) δ: 32.39 (*C*H_2_CO), 48.40 (NCH_2_), 105.18, 114.71, 123.06, 125.49, 125.53, 125.76, 126.10, 127.24, 127.55, 128.60, 132.37, 138.56, 148.77, 148.88, 170.98, 172.75 (C_Ar_, COOH).

Anal. Calcd. for C_19_H_16_F_3_N_3_O_2_S, %: C 56.01; H 3.96; N 10.31. Found, %: C 57.57; H 4.21; N 10.22.

3-[(4-Acetamidophenyl)(4-oxo-4,5-dihydro-1,3-thiazol-2-yl)amino]propanoic acid (**7**)

To a solution of thioureido acid **4** (8.44 g, 30 mmol) and sodium carbonate (9.02 g, 90 mmol) in water (35 mL), monochloroacetic acid (5.67 g, 60 mmol) was added, and the mixture was heated at reflux for 3 h. After completion of the reaction, the mixture was cooled down and acidified with acetic acid to pH 6. The formed precipitate was filtered off and washed with water.

Yield 6.75 g (70%), m. p. 133–134 °C (from ethanol).

IR (KBr), ν, cm^−1^: 3542–3253 (NH, OH), 1716, 1694, 1667 (3 C=O).

^1^H NMR (400 MHz, DMSO-*d_6_*) δ: 2.07 (s, 3H, CH_3_), 2.55 (t, *J* = 7.3 Hz, 2H, CH_2_CO), 3.91 (s, 2H, SCH_2_), 4.13 (t, *J* = 7.3 Hz, 2H, NCH_2_), 7.37 (d, *J* = 8.6 Hz, 2H, H_Ar_), 7.69 (d, *J* = 8.6 Hz, 2H, H_Ar_), 10.18 (s, 1H, NH); 12.38 (br. s, 1H, COOH).

^13^C NMR (101 MHz, DMSO-*d_6_*) δ: 24.07 (CH_3_), 31.94 (*C*H_2_CO), 40.59 (SCH_2_), 49.97 (NCH_2_), 119.69, 128.60, 134.62, 140.24, 168.73, 171.99, 183.44, 187.06 (C_Ar_, CH_3_*C=*O, COOH, NC=O).

Anal. Calcd. for C_14_H_15_N_3_O_4_S, %: C 52.33; H 4.70; N 13.08. Found, %: C 52.54; H 4.68; N 13.16.

General procedure for the preparation of 3-{(4-acetamidophenyl)[5-(phenylmethyliden)-4-oxo-4,5-dihydro-1,3-thiazol-2-yl]amino}propanoic acids **8a**–**d.**

To a solution of compound **7** (1.6 g, 5 mmol) and sodium carbonate (2.12 g, 20 mmol) in water (10 mL), the corresponding benzaldehyde (4.7 mmol) was added, and the mixture was heated at reflux for 3 h. Then, the mixture was cooled down and acidified with acetic acid to pH 6. The formed was filtered off and washed with water. The purification was performed by dissolving in aqueous sodium carbonate solution (5 g, 60 mL of water) and filtering and acidifying the filtrate with acetic acid to pH 6.

3-[(4-Acetamidophenyl)(5-benzylidene-4-oxo-4,5-dihydro-1,3-thiazol-2-yl)amino]propanoic acid (**8a**)

Yield 1.76 g (86%), m. p. 227–228 °C.

IR (KBr), ν, cm^−1^: 3419–3112 (NH, OH), 1673, 1599, 1685 (3 C=O).

^1^H NMR (400 MHz, DMSO-*d_6_*) δ: 2.09 (s, 3H, CH_3_), 2.31 (t, *J* = 7.8 Hz, 2H, CH_2_CO), 4.17 (t, *J* = 7.8 Hz, 2H, NCH_2_), 7.21–7.54 (m, 7H, H_Ar_), 7.60 (s, 1H, CH), 7.77 (d, *J* = 8.5 Hz, 2H, H_Ar_), 10.60 (s, 1H, NH).

^13^C NMR (101 MHz, DMSO-*d_6_*) δ: 24.05 (CH_3_), 35.10 (*C*H_2_CO), 52.24 (NCH_2_), 119.74, 128.68, 129.21, 129.39, 129.48, 129.67, 129.79, 133.79, 134.58, 140.54, 168.87, 173.35, 175.86, 179.69 (C_Ar_, CH_3_*C*O, COOH, NCO).

Anal. Calcd. for C_21_H_19_N_3_O_4_S, %: C 61.60; H 4.68; N 10.26. Found, %: C 61.35; H 4.55; N 10.33.

3-((4-Acetamidophenyl){5-[(4-fluorophenyl)methylidene]-4-oxo-4,5-dihydro-1,3-thiazol-2-yl}amino)propanoic acid (**8b**)

Yield 1.43 g (67%), m. p. 220–221 °C.

IR (KBr), ν, cm^−1^: 3305–3115 (NH, OH), 1670, 1532, 1595 (3 C=O).

^1^H NMR (700 MHz, DMSO-*d_6_*) δ: 2.09 (s, 3H, CH_3_), 2.40 (t, *J* = 7.6 Hz, 2H, CH_2_CO), 4.18 (t, *J* = 7.6 Hz, 2H, NCH_2_), 7.27 (t, *J* = 8.6 Hz, 3H, H_Ar_), 7.38–7.55 (m, 4H, H_Ar_), 7.61 (s, 1H, CH=), 7.75 (d, *J* = 8.4 Hz, 2H, H_Ar_), 10.47 (s, 1H, NH).

^13^C NMR (176 MHz, DMSO-*d_6_*) δ: 24.06 (CH_3_), 34.09 (*C*H_2_CO), 51.53 (NCH_2_), 116.23, 116.45, 119.77, 128.71, 128.88, 129.15, 129.18, 130.43, 131.72, 131.81, 134.49, 140.53, 161.22, 163.70, 168.86, 172.92, 175.96, 179.61 (C_Ar_, CH, CH_3_*C*O, COOH, NC=O).

Anal. Calcd. for C_21_H_18_FN_3_O_4_S, %: C 59.01; H 4.24; N 9.83. Found, %: C 51.35; H 4.11; N 11.89.

3-((4-Acetamidophenyl){5-[(4-chlorophenyl)methylidene]-4-oxo-4,5-dihydro-1,3-thiazol-2-yl}amino)propanoic acid (**8c**)

Yield 1.55 g (70%). m.p. 228–229 °C.

IR (KBr), ν, cm^−1^: 3283–3123 (NH, OH), 1710, 1653, 1683 (3 C=O).

^1^H NMR (400 MHz, DMSO-*d_6_*) δ: 2.09 (s, 3H, CH_3_), 2.55 (d, *J* = 7.4 Hz, 2H, CH_2_CO), 4.21 (t, *J* = 7.4 Hz, 2H, NCH_2_), 7.30–7.88 (m, 9H, H_Ar_ + CH), 10.32 (s, 1H, NH).

^13^C NMR (101 MHz, DMSO-*d_6_*) δ: 24.07 (CH_3_), 32.64 (*C*H_2_CO), 50.56 (NCH_2_), 119.75, 128.72, 128.88, 129.29, 129.98, 131.08, 132.62, 134.26, 140.54, 168.80, 172.21, 176.12, 179.45 (C_Ar_, CH_3_*C*O, COOH, NCO).

Anal. Calcd. for C_21_H_18_ClN_3_O_4_S, %: C 56.82; H 4.09; N 9.47.

Found, %: C 51.35; H 4.11; N 11.89.

3-((4-Acetamidophenyl){5-[(4-bromophenyl)methylidene]-4-oxo-4,5-dihydro-1,3-thiazol-2-yl}amino)propanoic acid (**8d**)

Yield 1.61 g (66%), m. p. 243–244 °C.

IR (KBr), ν, cm^−1^: 3279–3054 (NH, OH), 1708, 1653, 1683 (3 C=O).

^1^H NMR (400 MHz, DMSO-*d_6_*) δ: 2.09 (s, 3H, CH_3_), 2.55 (d, *J* = 7.4 Hz, 2H, CH_2_CO), 4.22 (t, *J* = 7.4 Hz, 2H, NCH_2_), 7.38 (d, *J* = 8.4 Hz, 2H, H_Ar_), 7.47 (d, *J* = 8.6 Hz, 2H, H_Ar_), 7.59 (s, 1H, CH), 7.63 (d, *J* = 8.4 Hz, 2H, H_Ar_), 7.74 (d, *J* = 8.6 Hz, 2H, H_Ar_), 10.30 (s, 1H, NH).

^13^C NMR (101 MHz, DMSO-*d_6_*) δ: 24.10 (CH_3_), 32.46 (*C*H_2_CO), 50.45 (NCH_2_), 119.82, 123.19, 128.76, 129.08, 130.08, 131.31, 132.27, 132.97, 134.29, 140.57, 168.89, 172.15, 176.22, 179.51 (C_Ar_, CH_3_*C*O, COOH, NCO).

Anal. Calcd. for C_21_H_18_BrN_3_O_4_S, %: C 51.65; H 3.72; N 8.60. Found, %: C 51.34; H 3.62; N 8.36.

3-((4-Aminophenyl)(4-(4-chlorobenzylidene)-5-oxo-4,5-dihydrothiazol-2-yl)amino)

propanoic acid (**9**)

Compound **8c** (0.25 g, 0.56 mmol) was dissolved in 5% hydrochloric acid 6 mL, and the obtained mixture was refluxed for 1 h. Then, the mixture was cooled down and neutralized with sodium acetate to pH 6. The formed precipitate was filtered off.

Yield 0.23 g (92%). m. p. 192–193 °C (from 2-propanol).

IR (KBr), ν, cm^−1^: 3401; 3050 (NH_2_, OH), 1735, 1591 (2 CO).

^1^H NMR (400 MHz, DMSO-*d_6_*) δ: 2.54 (t, *J* = 7,5 Hz, 2H, CH_2_CO), 4.15 (t, *J* = 7,6 Hz, 2H, NCH_2_), 5.61 (br. s, 2H, NH_2_), 6.64 (d, *J* = 8,2 Hz, 2H, H_Ar_), 7.12 (d, *J* = 8,2 Hz, 2H, H_Ar_), 7.48 (dd, *J* = 11,4, 5,0 Hz, 4H, H_Ar_), 7.59 (s, 1H, CH).

^13^C NMR (101 MHz, DMSO-*d_6_*) δ: 32.23 (*C*H_2_CO), 50.26 (NCH_2_), 114.01, 127.47, 128.44, 128.78, 129.29, 130.45, 131.02, 132.74, 134.14, 150.09, 168.79, 176.62, 179.58 (C_Ar_, COOH, NCO).

Anal. Calcd. for C_19_H_16_ClN_3_O_3_S, %: C 56.98; H 4.15; N 10.60.

Found, %: C 56.79; H 4.01; N 10.46.

1-(4-Aminophenyl)-2-thioxotetrahydropyrimidin-4(1*H*)-one (**10**)

Compound **3** (0.39 g, 1.5 mmol) was dissolved in 5% hydrochloric acid 15 mL, and the obtained mixture was boiled for 1 h. Then, the mixture was cooled down and neutralized with sodium acetate to pH 6. The formed precipitate was filtered off and washed with water.

Yield 0.23 g (70%). m. p. 219–220 °C (from 2-propanol).

IR (KBr), ν, cm^−1^: 3399; 3325 (NH_2_, NH); 1712 (CO).

^1^H NMR (400 MHz, DMSO-*d_6_*) δ: 2.75 (t, *J* = 6,9 Hz, 2H, CH_2_CO), 3.80 (t, *J* = 6,9, Hz, 2H, NCH_2_), 5.20 (br. s, 2H, NH_2_), 6.54 (d, *J* = 8,4 Hz, 2H, H_Ar_), 6.93 (d, *J* = 8,3 Hz, 2H, H_Ar_), 11,06 (br. s, 1H, NH).

^13^C NMR (101 MHz, DMSO-*d_6_*) δ: 32.46 (CH_2_CO), 50.45 (NCH_2_), 119.82, 123.19, 128.76, 129.08, 130.08, 131.31, 132.27, 132.97, 134.29, 140.57, 168.89, 172.15, 176.22, 179.51 (C_Ar_, CO, CS).

Anal. Calcd. for C_10_H_11_ClN_3_OS, %: C 54,28; H 5,01; N 18,99.

Found, %: C 54.23; H 4.99; N 18.89.

3-(1-(4-Aminophenyl)thioureido)propanoic acid (**11**)

Compound **9** (0.5 g, 2,26 mmol) was dissolved in hot, aqueous 5% sodium hydroxide (10 mL), and the obtained solution was filtered off. After cooling to room temperature, the solution was acidified with 5% HCl to pH 6. The formed solid was filtered off and washed with water.

Yield 0.47 g (87%). m. p. 171–172 °C (from 2-propanol).

IR (KBr), ν, cm^−1^: 3407; 3340; 3276 (2× NH_2_, OH); 1742 (CO).

^1^H NMR (400 MHz, DMSO-*d_6_*) δ: 2.48–2.50 (signal of the CH_2_CO (2H) overlaps with the DMSO-*d_6_*), 4.12 (t, *J* = 6.9, Hz, 2H, NCH_2_), 5.32 (br. s, 2H, NH_2_CS), 6.59 (d, *J* = 8.5 Hz, 2H, H_Ar_), 6.84 (d, *J* = 8.5 Hz, 2H, H_Ar_), 7.45 (br. s., 2H, NH_2_), 12.30 (br. s, 1H, OH).

^13^C NMR (101 MHz, DMSO-*d_6_*) δ: 32.26 (*C*H_2_CO), 50.55 (NCH_2_), 114.63, 128.02, 148.63, 172.59, 181.71 (C_Ar_, CO, CS).

Anal. Calcd. for C_10_H_13_N_3_O_2_S, %: C 50.29; H 5.48; N 17.56.

Found, %: C 50.21; H 5.45; N 17.56.

### 4.2. Cell lines and Culture Conditions

MDCK cells were kindly provided by Dr. Mirella Salvatore (Department of Medicine, Division of Infectious Diseases, Weill Cornell Medicine of Cornell University) and were maintained in Dulbecco’s MEM (Life Technologies, Burlington, ON, Canada) (DMEM) supplemented with 10% fetal blood serum (FBS), 100 U/mL penicillin, and 100 µg/mL streptomycin. A549 cells were obtained from American Type Culture Collection (ATCC, Manassas, VA, USA) and were cultured in Dulbecco’s Modified Eagle Medium/Nutrient Mixture F-12 media supplemented with 10% FBS, 100 U/mL penicillin, and 100 µg/mL streptomycin.

### 4.3. Cytotoxicity Assay

The cytotoxicity of compounds **3**–**11** on A549 human lung cells were evaluated by using a commercial MTT assay (CyQUANT MTT Cell Viability Assay, Thermo Fisher Scientific, Eugene, Oregon, USA). A549 was plated to flat-bottomed 96-well plates (1 × 10^4^ cells/well) and incubated overnight to facilitate the attachment. The compounds at fixed 100 µM concentration were added, and plates were further incubated for 48 h at 37 °C, 5% CO_2_. After incubation, the commercial MTT reagent was added, and the % of viability was determined, in accordance with the description of the manufacturer, using untreated cells as a control. All experiments were performed in triplicate.

### 4.4. Viral Infection Assay

To determine the potential antiviral activity of compounds **3**–**11** on the virus replication in MDCK cells, we used virus-induced cell death as an experimental output. Briefly, MDCK cells were plated to flat-bottomed 96-well plates (1 × 10^4^ cells/well) and incubated overnight to facilitate the attachment. After incubation, the media was removed, cells were gently washed twice with DPBS, and the compounds (100 µM) were dissolved in DMEM supplemented with 5% bovine serum albumin (BSA), 2 µg/mL TPCK-treated trypsin (Thermo Fisher Scientific, Rockford, Illinois, USA), 100 U/mL penicillin, and 100 µg/mL streptomycin. The oseltamivir and amantadine (100 µM) were used as a comparator. Cells were incubated with compounds for 1 h at 37 °C, 5% CO_2_, and were then infected with influenza A/Puerto Rico/8/34 H1N1 strain at MOI 1:5. The infected cells were then further incubated for 24 h to facilitate the infection, and the remaining post-infection viability was measured by using MTT assay.

### 4.5. Measurement of Antioxidant Activities

#### 4.5.1. Ferric ion (Fe^3+^) Reducing Antioxidant Power (Fe^3^ – Fe^2+^ Transformation Assay)

Newly synthesized compounds **3**–**11** of concentration 20 mM in 0.5 mL of DMSO were mixed with phosphate buffer (1.25 mL, 0.2 M, pH 6.6) and potassium ferricyanide [K_3_Fe(CN)_6_] (1.25 mL, 1%). The mixture was incubated at 50 °C for 20 min. Aliquots (1.25 mL) of trichloroacetic acid (10%) were added to the mixture, which was then centrifuged for 10 min at 9000 rpm. The upper layer of solution (1.25 mL) was mixed with distilled water (1.25 mL) and FeCl_3_ (0.25 mL, 0.1%), and the absorbance was measured at 700 nm in a spectrophotometer [47,48]. Butylhydroxytoluene (BHT) was used as a positive control.

#### 4.5.2. Ferric Reducing Antioxidant Power Assay (FRAP)

Reducing properties were investigated using the FRAP method, which is based on the reduction of a ferric-tripyridyl triazine complex to its ferrous-colored form in the presence of antioxidants [49]. The FRAP reagent contained 2.5 mL of 10 mM TPTZ (2,4,6-tripyridyl-*s*-triazine) solution in 40 mM HCl as well as 2.5 mL of FeCl_3_ (20 mM) and 25 mL of acetate buffer (0.3 M, pH = 3.6). Then, 100 μL of analyzed compounds (20 mM) were mixed with 3 mL of the FRAP reagent. The absorbance of the reaction mixture was measured spectrophotometrically at 593 nm. For comprising the calibration curve, five concentrations of FeSO_4_
**·** 7H_2_O (5, 10, 15, 20, 25 μM) were used, and the absorbance was measured as a sample solution. Each experiment was repeated three times.

#### 4.5.3. 1,1-Diphenyl-2-picrylhydrazyl (DPPH) Radical Scavenging Assay

The free-radical-scavenging activity of **3**–**11** compounds was measured by the DPPH method [48,50]. Firstly, a solution (20 mM) of **3**–**11** compounds was prepared in DMSO. Then, a 1 mM solution of DPPH in ethanol was prepared, and 1 mL of this solution was added to the solutions of the analyzed compounds. The mixture was vigorously stirred and allowed to stand at room temperature. After 20 min, the absorbance of the reaction mixture was measured at 517 nm with a UV-1280 spectrophotometer (Shimadzu, Kyoto, Japan). Each experiment was repeated three time.

### 4.6. Evaluation of Antibacterial Activity

Antibacterial activity of the compounds was screened by using the disk-diffusion method [51]. In this study, inhibition of bacterial growth was investigated against Gram-positive bacteria *Bacillus subtilis* and Gram-negative bacteria *Escherichia coli*. The solution (20 mM) of the compounds was prepared in DMSO. Bacterial cultures were cultivated in Petri dishes at 37 °C for 24 h on the Luria-Bertani (LB) agar medium. Then, 50 μL inoculum containing bacterial cells were spread across the LB agar medium. Sterile filter-paper disks were soaked in 25 μL of each compound solution, and then the disks were put on the LB agar medium. Ciprofloxacin (20 mM) was used as positive control, and DMSO was used as the negative control. Petri dishes were incubated aerobically at 37 °C and examined for zones of inhibition after 24 h. The inhibition zones (cm) were measured.

## 5. Conclusions

The obtained results revealed that selected aminothiazole compounds bearing 4-cyanophenyl, 4-chlorophenyl, and 4-trifluoromethylphenyl substituents in the thiazole ring could act, with a built-in capacity, on three targets, with the combination of a single structure with antiviral, antioxidant, and antibacterial activities. Among the synthesized compounds, 3-{(4-aminophenyl)[4-(4-cianophenyl)-1,3-thiazol-2-yl]amino}propanoic acid (**6d**) showed the highest cytotoxic properties. The selected compounds 3-((4-aminophenyl){4-[4-(trifluoromethyl)phenyl]thiazol-2-yl}amino)propanoic acid (**6e**) and 3-{(4-acetamidophenyl)[4-(4-trifluoromethylphenyl)-1,3-thiazol-2-yl]amino}propanoic acid (**5e**) exhibited significant influenza A-targeted antiviral activity. Future work can be focused on application of thiazoles scaffolded in bioluminescent systems to develop novel substrates for luciferase.

## Data Availability

Not applicable.

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
