# Peer review of "Synthesis of Novel Aminothiazole Derivatives as Promising Antiviral, Antioxidant and Antibacterial Candidates"

_ijms, 2022, doi:10.3390/ijms23147688_

Round 1
Reviewer 1 Report
The manuscript entitled “Synthesis of novel aminothiazole derivatives as promising antiviral, antioxidant and antibacterial candidates” by R. Minickaitė et al. described the synthesis of a series of novel substituted aminothiazoles and study their antiviral, antioxidant and antibacterial activities to find potential agents that could have biological activities in one single biomolecule. The manuscript may be of general interest to the researchers of this field, but the manuscript lacks some information that the author should consider and incorporate in the present form of the manuscript. Here are a few concerns that need to be addressed in the present form of the manuscript.
1. The information about antiviral model (line 31) should be added in the abstract.
2. It should be good to add conclusion as separate section.
3. The titles of items 2.2, 2.3 should be corrected. For example, ”Study of …… activity of compounds …”.
4. In the experimental part, 1H BMR and 13C BMR should be corrected on 1H NMR and 13C NMR (from line 381…).
5. The formula of compound 5b should contain subscripts (line 420).
6. The names of compounds should be capitalized (lines 374, 398, 410, 423, 435, 447).
7. In line 701 should be “FeSO4 · 7H2O”.
8. The full name of BHT should be added in the text.
9. Fig. 1 contain “DOX” on the horizontal scale "Compounds", but the description for figure contains cisplatin (CP).
10. What does UIC mean (Fig. 2)?
Author Response
Dear Reviewers,
Please find a new version of manuscript, which has been thoroughly revised addressing reviewers comments and recommendations. On behalf of all authors, we are thankful for the reviewers for their time and effort dedicated during reviewing this paper. All comments raised by the reviewers were fully addressed and the changes were incorporated in this manuscript. Please see the attachment. Sincerely, Ilona Jonuškienė

Reviewer 2 Report
The authors synthesized a set of novel aminothiazole derivatives and analyzed all of them to determine their antiviral, antibacterial and antioxidant capabilities.
The article is well written and follow the scientific method of investigation and presentation of the results.
In total, if I counted well, with all derivatives, there are 20 novel aminothiazole derivatives which were synthesized and their antiviral, antibacterial and antioxidant checked.
In my opinion the article can be published in the actual form, I only identified some minor spelling inaccuracies and I present them on the following:
Page 1 Line 23 - Already at the beginning of the Abstract: “It is well-known that thiazole derivatives usually found in lead structures, demonstrating a wide range of pharmacological effects.” it is not enough clear in my opinion. You can probably write: “It is well-known that thiazole derivatives ARE usually found in lead structures WHICH DEMONSTRATE (or WHICH PRESENT) a wide range of pharmacological effects.” And even if “lead structures” is an already settled term for compounds presenting the best performance/activity is a bit too vague in my opinion to start with it.
Page 4 Line 141-142: “analysisand” must have a space in between: “analysis and”
Page 4 Line 143-145: The following phrase is not very clear must be reviewed: “The condensation of the obtained thiazolone 7 with various aromatic aldehydes was performed under analogous conditions, like and synthesis of a thiazolinone 7, instead the monochloroacetic acid using the corresponding aromatic aldehydes.”
Firstly, compund 7 appear with two names “thiazolone 7” and “thizolinone 7”. Then “under analogous conditions, like and synthesis of a thiazolinone 7” can be written: “under analogous conditions with the synthesis” or “under analogous conditions, like the synthesis” and it will be better to use the preposition "of" in “instead the monochloroacetic acid” to become “instead of the monochloroacetic acid”
Page 4 Line 163: In the caption of Schema 3 “Synthesis compounds 10 and 11.” The preposition “of” was forgotten again.
Page 4 Line 164: In my opinion in spite of “product 10 was applied” you can better say “product 10 was used”
Page 4 Line 167-168: In the phrase “Comparison of the spectra of compounds 3 and 10 showed the singlet of the methyl group of the acetamide moiety is absent” you must say “showed that the singlet” and on the following “but broad singlet of an amino group is visible” you can say “but a broad singlet of an amino group is visible”.
Author Response
Dear Reviewers,
Please find a new version of the manuscript, which has been thoroughly revised addressing reviewers comments and recommendations. On behalf of all authors, we are thankful for the reviewers for their time and effort dedicated during reviewing this paper. All comments raised by the reviewers were fully addressed and the changes were incorporated in this manuscript.
Sincerely, Ilona Jonuškienė
